# Polymeric Heart Valves Will Displace Mechanical and Tissue Heart Valves: A New Era for the Medical Devices

**DOI:** 10.3390/ijms24043963

**Published:** 2023-02-16

**Authors:** Maria A. Rezvova, Kirill Y. Klyshnikov, Aleksander A. Gritskevich, Evgeny A. Ovcharenko

**Affiliations:** 1Research Institute for Complex Issues of Cardiovascular Diseases, 650002 Kemerovo, Russia; 2A. V. Vishnevsky National Medical Research Center of Surgery, 117997 Moscow, Russia

**Keywords:** biomaterials, durability, heart valve replacement, hemocompatibility, immune response, mechanical heart valves, nanocomposites, polymeric heart valves, tissue heart valves, transcatheter aortic valve implantation

## Abstract

The development of a novel artificial heart valve with outstanding durability and safety has remained a challenge since the first mechanical heart valve entered the market 65 years ago. Recent progress in high-molecular compounds opened new horizons in overcoming major drawbacks of mechanical and tissue heart valves (dysfunction and failure, tissue degradation, calcification, high immunogenic potential, and high risk of thrombosis), providing new insights into the development of an ideal artificial heart valve. Polymeric heart valves can best mimic the tissue-level mechanical behavior of the native valves. This review summarizes the evolution of polymeric heart valves and the state-of-the-art approaches to their development, fabrication, and manufacturing. The review discusses the biocompatibility and durability testing of previously investigated polymeric materials and presents the most recent developments, including the first human clinical trials of LifePolymer. New promising functional polymers, nanocomposite biomaterials, and valve designs are discussed in terms of their potential application in the development of an ideal polymeric heart valve. The superiority and inferiority of nanocomposite and hybrid materials to non-modified polymers are reported. The review proposes several concepts potentially suitable to address the above-mentioned challenges arising in the R&D of polymeric heart valves from the properties, structure, and surface of polymeric materials. Additive manufacturing, nanotechnology, anisotropy control, machine learning, and advanced modeling tools have given the green light to set new directions for polymeric heart valves.

## 1. Introduction

Valvular heart disease (VHD) is an underestimated cause of disability, morbidity, reduced quality of life, and premature mortality [1]. In the general population, the age-adjusted prevalence of moderate or severe VHD comprised 2.5% [2], with the further prevalence increase from 0.7% in 18–44-year-olds to 13.3% in the 75 years and older group [3]. The marked undertreatment of patients with VHD is clearly apparent in the steady increase in the number of surgical and transcatheter interventions [4]. Currently, the most common treatment for VHD is surgical heart valve replacement with either mechanical or tissue valve prosthesis, with approximately 250,000–300,000 replacements globally per year [5]. Mechanical heart valves (MHV) are commonly implanted in 55% of cases, whereas tissue heart valves (THV) are chosen in 45% of cases [6]. The decision-making process on optimal heart valve substitutes is driven mainly by patients’ own choice, age, contraindications to anticoagulation therapy, and place of residence [7]. Despite the superior durability of MHV, high shear stresses may cause the activation of blood elements and initiation of platelet aggregation, causing thrombus formation. Therefore, MHV recipients need to take life-long anticoagulation therapy to prevent thromboembolic events. THV that mimic native valves do not require such therapy but have a rather limited lifespan due to structural valve degradation, including calcification (Figure 1) [8]. THV recipients will require reoperation in 13.4–36.6% of cases within five years after the index surgery [9]. Each reoperation is associated with a higher rate of perioperative complications and mortality due to the present comorbidities, advanced age, and surgical complexity itself [9]. Recently emerged minimally invasive percutaneous procedures, such as transcatheter aortic valve replacement (TAVR), have indicated a tendency towards a wider application of THV [10]. Transcatheter approaches are used either for the first heart valve replacement or for a redo «valve-in-valve» implantation [11]. All recent commercially available transcatheter systems have proved promising results, especially in the elderly group with contraindications to surgical aortic valve replacement [7,12]. However, the use of thinner pericardium and biomaterial micro-damages during crimping questions the long-term TAVR durability and freedom from SVD, taking into account the expansion of indications to lower- and moderate-risk patient groups [13,14].

Despite recent advances in the development of prosthetic heart valves, they obviously necessitate a new generation of heart valves that may address the formers’ major drawbacks. Biomaterials for the development of next-generation heart valve prostheses should mimic the mechanical and hemodynamic properties of native valves [15]. Therefore, tissue-engineered heart valves (TEHV) represent the best option for the treatment of VHD since the patient`s own cells are used in its development, providing superior biocompatibility [16]. Recent limitations, including poor control over the biodegradation process in the physiological environment, the formation of extracellular matrix, and inferior mechanical parameters of the neo-organ, can lead to rapid valve dysfunction and failure [17].

PHVs may overcome major drawbacks of tissue and mechanical prostheses since their recipients will require neither life-long anticoagulation therapy nor repeated heart valve replacement within 7–9 years. These benefits are of particular importance for young adults and elderly adults as they may significantly improve the quality of life and ensure superior clinical outcomes following the surgery. TAVR has proven to be a viable tool for the high-surgical-risk population and may become even more superior when biological material is replaced with elastic biostable polymer. Polymeric heart valves (PHV) are able to provide superior mechanical strength and fatigue resistance along with required flexibility, biocompatibility, and calcification resistance. PHV can be implanted in patients of any age group compared to THV [15]. A variety of biocompatible and biostable polymers can be used for polymeric valves [18,19,20,21]. PHVs are superior to THVs in terms of the absence of antigens (e.g., galactose-alpha-1,3-galactose and N-glycolylneuramic acid) that are known to trigger the immune response in tissues [22]. Moreover, synthetic materials completely negate the risk of developing spongiform encephalopathy, which persists in cattle-derived tissues [23]. The development of flexible polymeric leaflets should ease the fabrication and manufacturing process (compared to THV) via high-throughput reproducible techniques (e.g., injection and compression molding, dip molding, etc.) [18]. Previously mentioned advantages emphasize the relevance of developing an ideal PHV and assessing its safety and durability. This review provides the rationale for choosing optimal biocompatible material potentially suitable for developing the new-generation medical device.

Over the past decade, several studies have reported the successes and failures in the development of an ideal polymeric heart valve [15,18,24]. However, they are focused on describing the designs of PHVs with little attention paid to polymers, their properties, composition, and surface. In this review, we report recently developed materials, such as SiPUU (LifePolymer, Foldax, UT, USA) and the first in-human implantation of these PHVs [25], FGO-PCU (Hastalex) [26], a unique polymer with incorporated graphene, SIBS nanocomposites [21], and other block copolymer materials. In addition, all relevant information on the polymer structure, properties, and the design of the prosthesis is summarized along with in vitro and in vivo testing results. In addition, previous reviews do not consider, in detail, the relationship between the chemical structure of polymers and the valve’s durability and functioning. Several sections of this review discuss the impact of the polymeric structure, composition, and surface on the end-use mechanical properties, hemocompatibility, calcification, and biocompatibility. Major drawbacks, including pannus, calcification, biodegradation, and thrombotic complications, are highlighted with fundamental mechanisms provoking these complications. Key trends, ideas, and technologies (additive manufacturing, nanotechnologies, machine learning, anisotropy control, advanced modeling tools, etc.) that are able to set new directions for developing polymeric heart valves are discussed.

## 2. Historical Background

### 2.1. The First-Generation Polymers of Heart Valves

#### 2.1.1. Polysiloxanes

In the 1950s, Roe, Owsley, and Boudoures were the first to introduce the PHV concept [27]. The first polymers that attracted scientists’ attention were polysiloxanes, commonly referred to as silicones or silicone rubbers [27,28]. By that time, polysiloxanes (poly(dimethylsiloxane)s, PDMS) had already been successfully used as an antithrombogenic coating for needles, syringes, parts of drainage systems, catheters, dialysis shunts, artificial urethra, and bile duct repair [29]. Silicones are polymers with a silicon–oxygen backbone with various functional groups bonded to silicon atoms. They are semi-organic compounds by its nature (Figure 2) [30]. The high bonding energy of Si–O ensures the thermal and chemical stability of silicone materials. The mechanical strength of polysiloxanes depends on the chain length, side groups, and crosslinking extent [30]. Despite relatively good biocompatibility, flexibility, and fatigue resistance, the first clinical study failed due to a high mortality rate and postoperative thrombotic complications [28]. Despite the first unsuccessful experience, there were numerous attempts to fabricate polymeric heart valves using silicone elastomers [31,32]. Thus, in the 60s, 18 patients received a tricuspid aortic prosthesis based on Dacron and polysiloxane that was made by Roe [31,33]. The longest survival period was 33–61 months for four patients. Trileaflet polymeric heart valves made by Hufnagel were fabricated using composite silicone rubber–polypropylene fabric and implanted in 20 patients [32]. Of the 12 patients who survived for more than six months after surgery, 11 died due to structural valve degeneration or thrombosis [32]. Despite the wide use of polydimethylsiloxane-based devices and implants in different fields of medicine, this material could not withstand the pressure loading created by blood flow and resulted in structural valve degeneration.

#### 2.1.2. Polytetrafluoroethylene

Along with silicone heart valves, PHVs were developed using another promising polymer compound—polytetrafluoroethylene (PTFE). The chemical structure of PTFE, formed by strong carbon–carbon bonds and stable carbon–fluorine bonds (Figure 2), exhibits high bio-inertia and stability and a low coefficient of friction and surface tension [34]. Due to these unique properties, PTFE is used in various fields of medicine, including cardiovascular surgery [35]. PTFE vascular grafts are well-known and widely used medical devices, but PTFE was originally used to manufacture the leaflets for the first prosthetic heart valve. In 1961–1963, Braunwald et al. replaced the aortic valve in 23 patients using a fully polymeric heart valve made from PTFE [36]. The clinical application of PTFE valves resulted in high mortality, regurgitation, leaflet tears, complete shredding, and disruption in some cases [36]. In 1969, Robert W. Gore proposed a new method of producing fluoroplast based on heating and stretching, allowing them to achieve a highly porous, air-permeable (70%) structure. The material is formally known as the generic term expanded PTFE (ePTFE). Its discovery led to a new direction in the medical device market [37]. Improved mechanical properties of ePTFE allowed for widening its application and resuming the development of elastic polymeric heart valves [35]. In spite of the first preclinical trials reporting high rates of calcification [38], ePTFE was successfully used for manufacturing valved conduits for pulmonary valve replacements in pediatric patients undergoing the Ross procedure in Japan [39].

Miyazaki et al. reported an overall survival rate of 96.1% at 5 years, 95.3% at 10 years, and 95.3% at 15 years, and a freedom from conduit replacement of 96.3% at 5 years, 87.4% of 10 years, and 84.2% of 15 years in 902 pediatric patients who underwent right ventricular outflow reconstruction with pulmonary valve replacement [40]. According to researchers from Korea, the conduit with the bicuspid ePTFE valve demonstrated good long-term durability, but along with a gradual increase of the pressure gradient. An increase in the pressure gradient was reported to be the main reason for the valve dysfunction, requiring reoperation [41].

EPTFE bicuspid pulmonary valves with a leaflet thickness of 0.6 mm and 0.1 mm and of various porosity were implanted in the period from 2000 to 2009 [42]. The valves demonstrated adequate durability in the mid-term period (the longest follow-up period was 8.3 years). The authors noted that the use of a 0.1 mm non-porous ePTFE prevented the penetration of cells and leaflet thickening, demonstrating improved mobility, the flexibility of the leaflets, and lower transvalvular pressure gradients compared to the valves with an ePTFE leaflet thickness of 0.6 mm. The last failed in some cases due to stiffness and calcification. Despite the relatively successful results, the authors argue that this valve is not ideal for the right ventricular outflow tract reconstruction due to significant drawbacks, such as poor long-term durability [42].

Tricuspid ePTFE conduits were implanted in the pulmonary position and in the aortic position in children. Two types of ePTFE devices with leaflet thicknesses of 0.1 mm and 0.04 mm were evaluated [43]. The authors concluded that conduits with thin (0.04 mm) ePTFE leaflets were safe and reliable in mid-term follow-up and demonstrated an improved hemodynamic function of small-diameter implants in contrast to large-sized conduits (22 and 24 mm). Nevertheless, further accumulation of clinical data and analysis of explanted samples is required to assess the durability of ePTFE [43].

Transcatheter tricuspid valves with 0.1 mm ePTFE leaflets coated with phosphorylcholine were implanted in eight sheep for four weeks [44]. The leaflets remained thin, flexible, and full-fledged. There were no blood clots or calcific deposits. There was no fibrous tissue in the outflow portion of the ePTFE valve, whereas a small fibrous layer was observed in the lower parts of the leaflets in the inflow portion.

Despite promising mid-term outcomes, the application of ePTFE-valved conduits is currently limited to the pediatric population [45]. Medical devices made from ePTFE are not routinely used in other positions due to a high risk of dysfunction provoked by calcification, thickening, or tightening of the valves and poor hemodynamic performance.

#### 2.1.3. Polyurethane

Polyurethanes (PUs) are other polymers of choice potentially suitable for developing polymeric heart valve leaflets. PUs are a versatile class of polymers that can be synthesized through the reaction of di- or polyisocyanates (hard segments) with di- or polyols (soft segments) via polymerization (Figure 2). By changing the ratio of soft and hard segments, it is possible to obtain a polymer with various properties. Its excellent mechanical properties and high biocompatibility allow for considering polyurethanes as one of the most promising biomaterials.

The first successful implantation of a mitral valve made of flexible polyurethane with Teflon chordae tendineae in a human was performed in 1960 [46,47]. A 44-year-old woman was discharged from the hospital. She suddenly died four months after the operation, presumably from arrhythmia [47]. The authors of early studies of PU-based heart valves reported the susceptibility of the material to biodegradation, calcification, and thrombosis [48,49]. As a rule, medical PUs are based on polyesters and polyols [50]. When the ether bonds are oxidized, cellular processes decompose PUs via hydrolysis [51]. Such degradation eventually leads to prostheses dysfunction due to the loss of tensile strength as a result of surface cracking and loss of molecular weight [51].

Subsequently, various PUs were tested in animal trials [52,53]. In 1991, Jansen et al. developed an aliphatic polycarbonate urethane (PCU) valve with the leaflets formed on a stretched stent to minimize stresses during leaflet opening and closing [54]. An in vitro lifetime of 400–650 million cycles was reported by the end of the accelerated fatigue tests. Despite rather promising bench testing, animal trials revealed poor resistance to calcification and thrombus formation [55]. In 1994, the Leeds group designed an alpharabola valve made from PCU with the smallest radius at the center of the leaflet at the free edge and good opening characteristics [56]. In 1996, another valve design was proposed based on three PU leaflets fixed from the inside of the flexible PU frame, ensuring the closed-leaflet geometry, elliptical in the radial direction and hyperbolic in the circumferential direction. The testing reported rather good durability with a lifespan equivalent of 10 years [57].

In 2003, huge progress in PCU application was reported by ADIAM Life Science AG, which produced PCU heart valves for the mitral and aortic positions with the trade name ADIAM. Structural elements of both prostheses were made of multilayered PCU, cohesively bonded, but not glued material. Both prostheses were designed to mimic physiological flow [58]. The bench testing reported the in vitro durability of the mitral valve of up to 1 billion cycles representing 15.8 to 26 years of average human function [58], whereas aortic prosthesis demonstrated 300 million cycles representing 7.9 years of average human function. In vivo testing of the ADIAM mitral valve prosthesis using a calf animal model reported the absence of paravalvular leaks with minor calcific deposits close to the commissures. However, in vivo testing of the aortic valve demonstrated a rather high rate of subtotal obstructions and mild-to-moderate calcification close to the commissures. Therefore, a second in vivo study was required to confirm the advantages of the ADIAM valves [58]. However, both prostheses did not transit to the clinical trial phase since that time.

Despite satisfactory mechanical properties, poor biocompatibility and biostability limit the use of polyurethanes for developing an optimal PHV.

### 2.2. Causes of Failure of the First-Generation Polymeric Valves

Final mechanical properties and bio- and hemocompatibility directly depend on polymer molecules. Therefore, it is important to consider each factor causing damage in conjunction with the polymer chain.

#### 2.2.1. Mechanical Degeneration

Prolonged mechanical pressure overload is one of the main factors contributing to prosthesis dysfunction. The potential suitability of the biomaterial may be derived based on its mechanical properties and fatigue resistance. Various polymeric materials, such as poly(methyl methacrylate), polyetheretherketone, and polyvinylidene fluoride, with good biocompatibility and biostability, cannot be used for heart valve prosthesis due to their non-compliance with the requirements for elasticity and strength.

The pressure loading experienced by the leaflets is up to 1 MPa [18,59]. At the same time, native pulmonary and aortic leaflets are able to withstand the stress in the circumferential direction equal to 2.78 ± 1.05 and 1.74 ± 0.29 MPa, respectively [60]. Native tissues have a significant margin of safety and are capable of recovery and renewal during a person’s lifetime. Moreover, synthetic materials should have a large margin of safety, considering their inability to regenerate. Commercially available THVs are commonly made from animal pericardium that can withstand a tensile stress of 10 to 30 MPa, depending on the method of chemical modification [61,62]. The mechanical strength of an ideal synthetic polymer material should not be inferior to biological tissues. The strength of biomaterial is determined by the density and direction of the collagen fibers [18,63]. The chemical chain and its length (molecular weight), polydispersity, as well as manufacturing method (e.g., solution casting and melt compounding, electrospinning) should be considered for evaluating synthetic polymers [64,65]. Polymer materials with maximum stress ranging from 5 MPa for silicones to 22 MPa and even higher for ePTFE were successfully obtained, as shown by the findings of the previous studies [18,66,67]. The molecular weight of the studied polymers reached 60,000–100,000 kDa, and the main polymer chain was formed by polyolefins and polyesters containing various functional groups. The strength of the material depends on the molecular weight distribution; the narrower it is, the better the end-use properties [18]. Based on the literature data, the valve destruction of previous prostheses was caused by a combination of factors, and the initial low strength was not the main one.

In addition to high tensile strength, PHV leaflets should have moderate rigidity for opening and moderate elasticity for closing to prevent the reverse of blood flow. Moreover, PHV leaflets should be able to revert back to their previous state as a result of deformation under pressure loading, similar to native tissues.

Elastin is the first to be subjected to pressure loading in native valves. Upon further stretching, convoluted collagen fibers are straightened [68]. After that, the collagen fibers begin to resist deformation, and in the absence of tension, elastin returns them to a convoluted state [68,69]. The elasticity of synthetic polymer materials is ensured by chain straightening and the motion of macromolecules relative to each other, as well as the ability of the molecule to assume a more energetically favorable state [70]. Polymer macromolecules consisting of carbon and hydrogen atoms exhibit the highest elongation at break under loading due to the weak interaction of the -CH_x_- groups with each other (rubbers, polyisobutylene, polyethylene). The potential barrier in such molecules is not very far advanced [71,72]. An increase in the intra- and intermolecular forces emerging due to polar functional groups in the structure of a macromolecule or three-dimensional crosslinking increases the rigidity and reduces the plasticity of the material. A high molecular weight increases the degree of entanglement of the molecules, enabling the material to be pulled apart further before the chains break [70]. However, not every deformation is reversible. In an energetically favorable state, polymers change their shape irreversibly (plastic deformation) [73]. The boundary of plastic deformation depends on the properties of the material itself, the nature of the applied speed and force, as well as its duration. This type of deformation can lead to irreversible changes in the valve resulting in subsequent dysfunction and failure.

The abovementioned parameters characterize the material only from the point of view of single-pressure loading, while prosthetic heart valve dysfunction occurs more often due to long-term cyclic loading [18,74]. Despite the absence of constant stress in the valve leaflets, it has been shown that physiological loading–unloading cycles can have a cumulative effect leading to elongation of the material or residual deformation [49] as a result of viscoelastic creep and/or hysteresis of the polymer [75,76]. A rapid cycle change followed by a new load limits the material’s ability to regain its original shape and results in straining [49]. Elongation is accompanied by local thinning, which then leads to increased stresses and reduced fatigue resistance [49]. Low-molecular amorphous hydrogenated carbon polymers, such as polyethylene, polyisobutylene, and PDMS, are most susceptible to irreversible elongation due to weak intermolecular bonds [75,76,77]. The introduction of functional groups would increase the molecular weight and stiffness of the chains and lead to a better creep resistance of the material and a decrease in hysteresis [75,78].

Despite the fact that polymers used for manufacturing the first PHVs had adequate tensile strength, elasticity, and Young’s modulus, their properties did not match the long-term cycling loading, and as a result, the prostheses failed due to structural degeneration. Thus, the durability of silicone valves was limited by the mechanical properties of the polymer [31].

#### 2.2.2. Polymer Degradation under Physiological Conditions

Structural valve degeneration is associated with biodegradation provoked by the changes in the chemical structure of materials contacting the human body [79].

Following the implantation of the device, polymer surfaces quickly adsorb blood proteins as well as water, ions, proteins, and lipids. The absorption of different elements can lead to plasticization and changes in size and mechanical properties. These processes proceed even more rapidly in porous amorphous materials compared to crystalline or monolithic materials. Blood components adhere to the surface and initiate oxidative processes and/or hydrolysis [80,81,82]. Within a few days or weeks, a fibrous capsule forms around the device, and the rate of chemical release from activated cells gets noticeably reduced.

The chemical structure and morphology of the polymer determine its tendency to hydrolysis and oxidation [83]. Polymer biomaterials that contain functional groups consisting of carbonyls within heterochain (O, N, S)—esters, amides, imides, urethanes, carbonates, anhydrides, etc., are the most susceptible to hydrolysis. Some of these compounds, namely urethanes, imides, and amides, can be stable in vivo if the main polymer chain is hydrophobic and/or crystalline [81]. An example of the first-generation polyurethanes used for biomedical purposes prone to hydrolysis is polyester urethanes joined by ester linkages [51]. It is generally believed that α-2-macroglobulin, cholesterol esterase, phospholipase A2, protease K, and cathepsin play a key role in the biodegradation of polyurethanes, as well as neutrophils and monocytes, which secrete hypochlorous acid and lysosomal hydrolases upon reacting to foreign surfaces [51]. PCUs solve the problems of hydrolysis and oxidation of polyester urethanes [84]. The most resistant to hydrolysis are simple linear hydrocarbons and their derivatives, such as polyethylene, polypropylene, polyisobutylene, polystyrene, and fluoroplastics [82]. PDMS are also regarded as stable, although they may be subjected to hydrolysis under prolonged loading [85,86].

Polymers implanted into the human body are oxidized by peroxides produced by FBGC and macrophages, which are released near the implants in order to damage the biomaterial surfaces [87]. Sites favored for initial oxidation enable the cleavage of an atom or ion, providing stabilization of the resulting radical or ion. Chemical compounds that are subjected to oxidation are polyolefins (-CH_2_-*CH_2_-CH_2_-), vinyl polymers (-CH_2_-*CHR-CH_2_), polyethers (-CH_2_-O-*CH_2_-), and polyamines (-CH_2_-*N-CH_2_-). On the contrary, fluoropolymers (-CF_2_-CF_2_-CF_2_-), polyesters (-CH(CH_3_)-C(=O)O), and silicon (-Si(CH_3_)_2_-O-Si(CH_3_)_2_-O-) are resistant to oxidation [87].

Factors accelerating hydrolysis and oxidation of biopolymers include a high number of groups susceptible to the abovementioned processes in the structure of the macromolecule; the presence of hydrophilic functional groups [86] and amorphousness (low degree of crystallinity); and large surface area of the polymer [88]. These factors are critical for fabricating ideal polymeric heart valves [81]. A reduction in the biodegradation rate can be achieved by introducing hydrophobic groups to the macromolecule; additional three-dimensional crosslinking of polymer chains; an increase in the degree of crystallinity of the polymer via thermal annealing or orientation of macromolecules [89]; and a decrease in the total surface area of the material (for example, by using casting instead of electrospinning) [88]. High molecular weight increases the resistance of the polymer to the human body environment since it requires much more action on the macromolecular chain to affect the mechanical properties of the material [83,90].

Biological degradation of the prosthesis resulting in structural changes of the early polymeric heart valves eventually led to their complete failure, as happened, for example, with one of the first PU-based polymeric valves implanted in 1960 [47,48].

#### 2.2.3. Thrombotic Complications

In addition to direct damage to the material of the implantable device (mechanical degeneration or biodegradation), cardiac dysfunction in patients with a prosthetic valve can be caused by the formation of blood clots and subsequent thrombosis [91]. Thrombotic complications are provoked by the alterations in the hydrodynamics or the chemical nature of the polymer. Non-physiological flow behavior due to the high rigidity of the material or inadequate valve design (mainly for mechanical valves) can lead to high shear stresses contributing to cell membrane rupture, hemolysis, platelet activation, and thrombus formation [92,93,94,95]. Nevertheless, the use of flexible polymer materials and tricuspid leaflet design improves hemodynamics, reduces platelet activation with a thrombus formation, and minimizes damage to red blood cells [94,96].

The chemical composition and surface structure of the polymer also significantly affect the blood components. As a result of the blood–biomaterial interaction, blood proteins spontaneously adsorb to the biomaterial surface. The nature of adsorption depends on the adsorbing surface: hydrophobicity/hydrophilicity, electrostatic charge and chemical reactivity, and polymer surface topography [97]. In particular, it was found that the non-specific adsorption of proteins increases along with an increase in surface pitting due to a bigger contact area [98]. Hydrophobicity due to the presence of aliphatic -CH_2_-CH_3_, aromatic groups, and perfluorinated groups -(C_2_F_4_) is the other reason for the increased protein adsorption. Macromolecules with ionic groups (acidic or basic) and polar chemical groups (ester R-COOR, ether -C-O-C-, alcohol C-OH), on the contrary, reduce protein adsorption on the surface of the biomaterial [99,100]. Fibrin adsorbed on the surface of the biomaterial stimulates adhesion and platelet activation, and adsorbed blood clotting factors (in particular, factor XII) trigger the coagulation cascade. These processes, along with other events, lead to thrombosis [101,102].

The problem of thrombosis in polymeric heart valves is still important and relevant since almost all synthetic materials are thrombogenic to different extents [102].

#### 2.2.4. Calcification

Prosthetic heart valve failure may also be caused by calcification. Calcification is a pathological process involving the formation of insoluble calcium salts (calcium deposits) [103], which limit the mobility of the leaflets due to loss of elasticity or rupture [104]. There are several concepts explaining the calcification of polymers. Previously, the researchers suggested that the calcification of the PU-based leaflets occurred due to the adsorption of the cellular component and platelets on the surface of the material [105]. Other authors proposed that calcium deposits formed in biological prostheses were more often localized in bends and surface cracks in the material of polymeric leaflets [106]. In the case of segmented PU, cyclic loading led to the stretching of soft segments and loosening, where blood proteins and/or phospholipids passively accumulated, attracting calcium ions [107,108]. Polymer materials releasing low molecular weight compounds can also accelerate mineralization processes [108]. Moreover, local inflammation contributes to the progression of dystrophic calcification of native valves [109], which could be extrapolated to polymer prostheses, but this concept requires further study.

Our own research, consistent with the literature data, showed that ePTFE is susceptible to calcification in vivo, which is probably due to its porous structure and inflammatory response since calcific deposits were found in the pores [19].

#### 2.2.5. Pannus Formation

In addition to calcification, the inflammatory response to implanted polymer prosthesis can provoke the proliferation of connective tissue, known as pannus. Daebritz et al. observed the formation of pannus despite the absence of calcification and without the involvement of leaflets in the implantation of PCU valves in calves [53]. The causes of pannus formation in polymer valves remain poorly understood. However, it is known that the physical properties of biomedical devices, such as the hemodynamic alterations (due to implantation of the device), can cause the activation of leukocytes and contribute to the release of many inflammatory molecules [110]. Since biomaterials do not stimulate an adaptive immune response, lymphocytes can contribute by activating monocytes (macrophages) [110]. Polymer surfaces can also activate the complement system, a complex of proteins that promote inflammation. The adsorption of various proteins of the complement system, such as C3b, factor D, and factor H, is an important part of this process that triggers the complement cascade. At the same time, the activation of the complement system is influenced by the chemical nature of the biomaterial. Functional groups -NH_2_, -OH, and -COOH are able to covalently bind the C3b protein [111]. Moreover, the researchers have found that the C3 protein adsorbed on a foreign surface can generate C3 convertase since the conformationally altered C3 mimics the configuration of the bound C3b [111].

#### 2.2.6. New-Generation Polymer Materials

An ideal polymer material for PHV should meet the following requirements: (1) high molecular weight, narrow polydispersity (low content of low molecular weight component); (2) soft and hard segments in the macromolecule structure; (3) reliable packaging of polymer blocks preventing soft segments from creeping; (4) the absence of functional groups affected by hydrolysis, oxidation, and enzymatic processes or a special packaging of macromolecules limiting the access to such groups; (5) limited protein adhesion; (6) the absence of negative effects on the blood components (platelets, erythrocytes, cells); (7) the possibility of processing the material by traditional methods, such as solution casting, melting, etc. (molecular weight limit); and (8) economic accessibility.

Nevertheless, it is quite difficult to find a polymer that would meet all the presented criteria. Since certain properties (e.g., biocompatibility and durability) may vary based on the polymer compounds, it is possible to improve them by using fillers and various modifiers. The progress made from developing the first polymeric valves in the 1950s to the 2000s has underlain the development of novel biostable polymers that may give new insights into the development of PHV.

## 3. Novel Polymers: Material Synthesis and Properties

While previous reviews describe some of the current developments, this section highlights new discoveries in this field, including novel polymers, their unique properties, and valve designs for both surgical and transcatheter implantation.

Under-studied requirements for PHVs and properties of polymers limited the development of an ideal heart valve prosthesis. A low number of methods for the preclinical evaluation of biomaterials and a poor understanding of blood–biomaterial interaction also hindered progress in this field for a long time. The success of modern methods of fabrication and modification of polymers, as well as the development of new techniques like nanotechnology, contributed to the emergence of biomaterials with unique properties. Segmented copolymers and polymer nanocomposites are quite relevant today. This was prompted, among other things, by studies of the macro- and microstructure [68,112,113] and biomechanics [68] of native valves. We have identified six of the most promising developments of the last decade:(1)Polyhedral oligomeric silsesquioxane poly(carbonate–urea) urethane (POSS-PCU);(2)Nanocomposite graphene–PCU polymer (FGO-PCU, Hastalex);(3)Siloxane poly(urethane–urea) (SiPUU, LifePolymer);(4)Poly(styrene-b-isobutylene-b-styrene) (SIBS) and poly(styrene-b-4-vinylbenzocyclobutene-b-isobutylene-b-styrene-b-4-vinylbenzocylcobutene) (xSIBS);(5)Nanocomposite polyvinyl alcohol (PVA) and bacterial cellulose (PVA-BC);(6)Linear-low-density polyethylene (LLDPE) and hyaluronan-enhanced linear-low-density polyethylene (HA-LLDPE).

These materials differ by structure from the first-generation polymers and provide superior mechanical properties and biocompatibility. Table 1 presents the chemical structure and some properties of the above-mentioned polymers.

It should be noted that researchers around the world, trying to find the perfect polymer material for prosthetic heart valves, are not limited to “classic” stented valves for surgical open-heart valve replacement (SVR) and are studying the applicability of materials for a more relevant direction—transcatheter valve replacement (TVR). In this paper, we will not discuss data separately since the required mechanical properties and biocompatibility for SVR and TVR valves are largely similar and are determined less by the method of implantation and more by the lifespan.

### 3.1. POSS-PCU

High resistance of PCU to cyclic loading but poor biodegradation and calcification (see Section 2.1) forced the researchers to improve the properties of this polymer by creating composites based on polydimethylsiloxane and polyhedral oligomeric silsequioxane nanoparticles (POSS) [114]. These modified materials have demonstrated excellent properties and potential for PHV fabrication (Table 1).

In 2005, a research group at the University College London developed a POSS-PCU nanocomposite polymer (Si_8_O_12_) covalently bonded as a suspended block with a polymer molecule [129,130]. The authors suggested that POSS nanoparticles are integrated with both soft and hard PCU segments, forming a single elastic matrix [131], which allows the nanocomposite to be resistant to hydrolysis, oxidation, and calcification [132]. POSS improved tensile strength (up to 53.6 ± 3.4 N mm^−2^ vs. 33.8 ± 2.1 N mm^−2^ in PCU, *p* < 0.01) and hydrophobicity of the resultant material [114]. In vitro studies showed a significant increase in the biocompatibility of POSS-PCU compared to polyurethanes due to a decrease in platelet adhesion and calcification [133].

The design of SVR valves based on POSS-PCU is represented by a “classic” tricuspid design with a polymer stent comprised of titanium wire as an additional reinforcing circuit (Figure 3). The POSS-PCU valve was designed for both intra- and supraannular implantation in the aortic position [116]. Moreover, A. Seifalian’s group optimized the manufacturing process of the POSS-PCU valves, opting for a dip coating process consisting of repeated dipping of a metal mandrel with the stent into the polyurethane solution. The automated fabrication process enabled precise control over the thickness of leaflets. After drying at 60 °C for one hour, the free edges were trimmed and the leaflet thickness was measured at five locations across each leaflet using digimatic micrometer regions [116].

TVR valves made from biological tissues have become a revolutionary method of treating aortic valve stenosis. However, with the spread of the disease among young patients, long-term durability and thrombosis have become challenging. From this point of view, the development of polymeric valves allows for receiving more durable valves with negligible side effects and variable thickness of leaflets.

The TRISKELE transcatheter valve made from POSS-PCU was a trileaflet valve. The valve was comprised of a nitinol wire frame, leaflets, a sealing cuff made from POSS-PCU, and a skirt supporting the sealing cuff (Figure 3). The structural elements of the TRISKELE valve were manufactured separately. Leaflets and the sealing cuff were made from POSS-PCU by robot-assisted dip-coating of a stainless-steel mandrel into an 18% (*w*/*v*) polymer solution. This fabrication technology ensures the manufacturing of highly reproducible PHVs. The stent was made from nitinol by thermomechanical shaping of three sets of nitinol wires joined by crimping sleeves [134,135]. The mean thickness of the leaflets was 130 ± 10 µm. This approach allowed authors to obtain a single curvature for the open and closed state, reducing energy loss during the cycle. Another geometrical advantage of the valve involved a self-expanding nitinol frame with the outflow portion containing three lateral ribs protruding radially further than the flow-control structure. This geometrical feature reduced the impact on the surrounding tissues and dampened the pressure load transferred to the leaflets [135].

The TRISKELE valve underwent in vitro tests using a pulse duplicator system with a mock silicone aortic root mimicking native aortic pressures and blood flow. Three POSS-PCU valves of sizes 23, 26, and 29 were manufactured for the hydrodynamic assessment. The benchmark valves included the Edwards SAPIEN XT and the Medtronic CoreValve of the same sizes [135]. The TRISKELE valve slightly self-adjusted from the initial position in a pulsatile flow. However, physiological pressure and load were achieved. The POSS-PCU valves were performing similarly to controls during systole, excluding the tests with a small aortic root (21 mm). The device performance was characterized by higher levels of ∆P and smaller EOA compared to controls. Paravalvular regurgitation was minimal due to the improved geometry of the valve. The sealing cuff filled the gaps between the implant and surrounding tissue. The overall hydrodynamic performance of the POSS-PCU valves was considered satisfactory and was less energy-demanding. The controls exhibited higher ventricular workload during the cycle. Limitations such as minor self-adjustment in the flow, high levels of ∆P, and smaller EOA in the small root model could be addressed with further optimization of the valve geometry [135].

The TRISKELE valve underwent in vivo animal trials. The valve of size 26 was implanted (off-pump) in a sheep using a brachiocephalic approach in the orthotropic position with fluoroscopy-guided angiography. The animal was sacrificed 30 min after the implantation. The TRISKELE valve exhibited optimal hemodynamic performance after catheter removal (mean velocity 785 ± 159 mm/s, mean gradient 3.0 ± 1.3 mmHg, EOA 2.49 ± 0.46 cm^2^, peak velocity and peak gradient 1086 ± 275 mm/s and 4.5 ± 1.5 mmHg, respectively). The implanted device did not interfere with the coronary flow and ideally matched the left ventricular outflow tract. The delivery system ensured control over the deployment of the valve and was atraumatic and fully retrievable [136]. However, the presence of regurgitation in the animals may necessitate additional optimization in the valve design. Moreover, a short-term in vivo experiment did not address the question of TRISKELE’s ability to resist calcification and long-term functioning. These issues will probably be addressed in long-term animal trials.

### 3.2. SiPUU (LifePolymer)

SiPUU is an improved biostable elastomeric polyurethane commonly used in medical implants. This polymer is synthesized using a modified two-step solution polymerization method [118]. Soft segments are based on α,ω-bis(6-hydroxyethoxypropyl) poly(dimethylsiloxane) and urethane-linked poly(hexamethylene oxide) (PHMO) macrodiols at a ratio of 80:20 [137], whereas hard segments are composed of either 4,4′-methylenediphenyl diisocyanate (MDI) and a mixture of 1,3-bis(4hydroxybutyl)-1,1,3,3-tetramethyldisiloxane (BHTD) and 1,2-ethylenediamine (EDA) in a 50:50 molar ratio [117]. Chemical structure modification results in an increase in Young’s modulus and tear and tensile strength (Table 1). The SiPUU with macrodiols affects the structure of the hard segments, unlike traditional PU. The lack of well-organized hard segments is compensated by increased intermolecular hydrogen bonding and strengthening of the interfacial regions, thus preventing plastic deformation [117]. In vitro tests have shown that the new polymer is resistant to oxidation, which is similar to the polyurethane Elast-Eon 2A [138,139], used in clinical practice, which also contains PDMS segments in its structure. Thus, the presence of PDMS changes the structure of the polymer of the biomaterial, increasing resistance to the physiological environment.

The SiPUU polymer has been tested in accordance with the ISO 10993 standards [140]. Ex vivo testing on thrombogenicity has demonstrated almost undetectable levels of platelet adhesion and thrombus formation on polymer surfaces. Long-term implantation of a SiPUU valve prototype in sheep has shown no differences in thrombogenicity or tissue response compared to a widely used biological valve. The analysis of extractable and leachable compounds of SiPUU-based heart valves has a confirmed low toxic risk. The results of accelerated in vivo biostability corresponded to the data of the in vitro tests. The SiPUU polymer is a promising material for the development of fully synthetic heart valves [140].

The SiPUU (LifePolymer, Foldax, UT, USA) valves with the trade name “Tria Aortic Valve” and “Tria Mitral Valve” consist of a stent, two or three specifically designed leaflets (depending on the valve position), and a sewing cuff (Figure 3).

These synthetic prostheses are designed for aortic and mitral positions and have a sewing cuff at the base. The leaflets are seamlessly connected to the stent by using a technique described below [140]. The valves consist of leaflets solution-cast onto a polyetheretherketone (Zeniva PEEK, Solvay) stent. After casting, mechanical trimming is performed to finalize leaflet dimensions, and a PTFE felt sewing ring is secured onto the annular valve base.

The aortic valve “Tria” for open-heart surgery was tested in a sheep model [140]. Their properties were analyzed in comparison with the Edwards Perimount valve (Edwards Lifesciences, Irvine, CA, USA), selected as controls. Eight Tria valves (size 23 mm) were implanted into sheep. After 140 days, the Tria valves showed a lower pressure gradient compared to the controls. Long-term implantation revealed no differences in thrombogenicity or tissue response compared to a clinically-approved biological valve. The advantages of the polymer valve included better resistance to calcification and pannus formation compared to controls. These studies have shown that the new elastomer exhibits ideal properties for fully synthetic elastic heart valves.

In 2018, Steven J. Yakubov announced the development of a transcatheter “Tria” valve (manufactured by Foldax Inc.). The polymer leaflets and frame design have been optimized by FEA engineers to increase durability and reduce profile for low-risk TAVR patients. The transcatheter Tria is made using LifePolymer material (Figure 3). The valve has a self-expanding nitinol frame with leaflets and a sealing cuff of 10 mm. The transcatheter Tria valve of size 27 mm has passed bench tests and showed 100 million cycles in accelerated fatigue testing. The mean gradient was 3.6 mmHg, and EOA was 2.9 cm^2^. Tria TVR valve has also been studied in vivo [141]. The device was implanted in the aortic position with CPB. The valve was delivered through the ascending aorta and positioned to evaluate the effectiveness of the leaflets in vivo. The results showed deliverability, excellent hemodynamics, absence of surface fibrin, thrombosis or fibrosis after 30 days, good cuff healing, and clean leaflets at 90 days [141].

Foldax Inc. is the only company that has received FDA approval to assess the clinical efficacy and safety of the aortic heart valve “Tria” (ID ClinicalTrials.gov: NCT03851068). The authors implanted 15 valves in 15 patients to assess the primary safety and efficacy end-points (rate of adverse events, efficacy, hemodynamic parameters, and improvement of NYHA class) and to measure secondary end-points (stroke, transient ischemic attack, length of stay in the ICU) within the year after the implantation [25]. The first patient, 68 years old, suffered from dyspnea, NYHA class 2 heart failure, hypertension, hyperlimidemia, and asthma. On July 30, 2019, his heart valve was replaced with the “Tria” valve (diameter of 21 mm). Before the procedure, LVEF was 65%, the mean pressure gradient was 44 mmHg, and the peak pressure gradient was 74 mmHg. After 30 days of follow-up, LVEF improved by 5% (EF—70%), and the mean pressure gradient decreased to 19 mmHg. The second patient at the age of 50 years, who received a 23 mm Tria valve, had LVEF of 60–65%, a mean pressure gradient of 45 mmHg, and a peak pressure gradient of 79 mmHg at baseline. After 30 days of follow-up, LVEF improved to 70%, and the mean pressure gradient decreased to 10 mmHg. The clinical trial is still ongoing, but preliminary data are encouraging.

### 3.3. FGO-PCU (Hastalex)

Hastalex is a novel nanocomposite material based on the integration of amino-functionalized graphene oxide (FGO) nanomaterials into a novel PCU developed by Seifalian et al. [26]. FGO-based polymer received the trade name Hastalex. Polymer films are made by solution casting.

Preliminary testing has shown its superior ultimate tensile strength compared to ePTFE (57.1 MPa vs. 22.5 MPa), which substantially exceeds the load experienced by native valves. One of the reasons for high tensile strength is the presence of graphene oxide in the polymer matrix. The higher Young`s modulus of FGO-PCU (about 30 MPa) seems beneficial in terms of minimizing energy losses. The hydrodynamic performance of Hastalex valves will largely depend on the design of the prosthesis in general and leaflets in particular. However, the high elasticity of the polymer makes it possible to produce thinner leaflets, making it particularly promising for application in TVR. Meanwhile, mechanical properties testing using cyclic loading, hysteresis tests, and hydrodynamic tests are required to provide its overall benefits.

Lower contact angles compared to ePTFE (85.2 ± 1.1° vs. 127.1 ± 6.8°) ensure hemocompatibility and lower protein attachment onto the Hastalex surface. The cytotoxicity test reports high biocompatibility of FGO-PCU, along with high cell adhesion, viability, and proliferation. Thus, this novel nanocomposite is also more biocompatible according to cell proliferation and viability tests and less susceptible to calcification than ePTFE [26]. In vivo testing on rats has shown that Hastalex is resistant to calcification compared to ePTFE and cattle-derived pericardium, widely used in clinical practice.

As in the case of POSS-PCU, the introduction of carbon nanoparticles into the structure has improved the polymer base, suggesting its beneficial potential for application in cardiovascular surgery. Currently, only one prototype of a Hastalex-based polymer prosthetic heart valve has been developed, and there are no data on its hydrodynamic performance or in vivo testing [26].

### 3.4. SIBS and xSIBS

Several research groups and biomedical companies have been working on a novel polymer, now known as SIBS, since the late 1990s (Kennedy and Ivan, Akron, OH, USA; Corvita Company, Doral, FL, USA; Innovia LLC, Miami, FL, USA; Stony Brook, NY, USA) [142]. Originally developed by Kennedy et al., SIBS was first commercialized by Boston Scientific Corporation’s (BSC) (Natick, MA, USA) in the Drug Eluting TAXUSs Coronary Stent in 2002. Then, Innovia LLC tried using it for biomedical applications, particularly for manufacturing PHV. SIBS is a thermoforming triblock copolymer with a central block that comprises polyisobutylene and end blocks of polystyrene. SIBS has a microphase-separated morphology in which the polystyrene phase forms physical crosslinks in the matrix of the rubbery polyisobutylene phase [120]. Due to its chemical composition, SIBS imbricates both silicone rubbers and polyurethanes and is highly resistant to oxidation, hydrolysis, and enzymatic cleavage due to the absence of unprotected ester, amide, ether, carbamate, and urea in the polymeric backbone and pendant groups [120]. SIBS is synthesized by living cationic polymerization. The molecular weight of the triblock is controlled by reaction conditions, mainly by the ratio of monomers/initiators. The hardness of SIBS can be varied by the amount of styrene employed. However, this material has several drawbacks: susceptibility to stress cracking, poor creep properties [120], and fatigue failure [143]; thus, the main focus of the researcher is aimed at improving the physical properties of the material, mainly via fiber reinforcement. Studies focused on SIBS optimization included its reinforcement with polypropylene [143] and polyethylene terephthalate (PET, Dacron) fibers [144], resulting in improved tensile and fatigue properties. Fiber reinforcement improved the properties of the material, but in vivo tests were unsuccessful due to the destruction of both the “base” and the fiber [143]. As a result, the researchers attempted to modify neat SIBS. Modified by Innovia LLC, the polymer was known as xSIBS, a new polyolefin thermoset elastomer created to prevent creep and negate the need for Dacron reinforcement [24,145]. A coupling reaction catalyzed by benzocyclobutene causes polymer backbones to crosslink via a Diels–Alder reaction. These modifications improved the mechanical properties of xSIBS compared to neat SIBS. The former exhibits nonlinear hyper-elastic behavior under tensile loading with high breaking strength and a large elastic range, being flexible and tear-resistant [145]. Its superior properties open up possibilities to xSIBS of becoming potential material for SVR and TVR valves.

The SIBS valve design underwent several improvements to ensure proper hemodynamic performance and durability. Its profile was changed from low to medium. Original spherical leaflets were changed into elliptical ones and were chosen as the better option [143,146]. The original SIBS-Dacron PHV was made by dip coating Dacron in dissolved SIBS. However, dip coating is associated with several limitations, and the major one was incomplete encapsulation of the Dacron, which promoted platelet activation [147]. It necessitated the optimization of the fabrication process. The new method included the impregnation of Dacron mesh embedded in cast SIBS dissolved in 15% toluene. The leaflets had a uniform thickness and were cut and sewn onto styrene molded SIBS valve stent using polyester sutures. However, all these changes to geometry were not enough to develop clinically-approved heart valve prostheses. Studies on animals reported Dacron calcification, fatigue failure, and SIBS leaflet coating cracking, which required further modification of its geometry and fabrication methods [148].

The problem was partially solved by using xSIBS and optimized design [148]. The leaflet shape was changed from cylindrical to hemispherical (Figure 3). In particular, the optimization of the geometry and leaflets was performed using parametric structural FEA analysis containing a series of simulations for the original Innovia valve and a benchmark biological valve. As a result, the leaflets’ thickness profile was adjusted locally to minimize the stresses during the cardiac cycle; regions of high flexural stresses were thickened, and vice versa. A flat leaflet profile ensured the maximum coaptation surface. The optimized PHV required further changes in the valve stent regarding its edges, widening the outflow portion by 2 mm (from 19 mm to a 21 mm internal diameter), thus maximizing the coaptation surface for optimal load distribution during closure [148]. Altering key parameters allowed Claiborne et al. to develop a hemodynamically optimized PHV potentially suitable for a clinical application that was later called the first-generation Polynova surgical aortic valve.

According to thrombogenicity assessment and further optimization, a one-step custom compression mold with special geometries was used. The homogenous xSIBS was chosen as an optimal polymeric material. The mold was covered with dry Teflon mold release agents, and then the mold was filled with raw 23% styrene xSIBS, sandwiched between water-cooled heating platens set to 260 °C, and compressed with 1 ton of force for 30 min. Air and excess polymer were extruded through portholes. The valve was removed using 200-proof ethanol, followed by crosslinking check by soaking the valve in toluene.

Claiborne et al. developed a SIBS-Dacron trileaflet valve mounted on a nitinol self-expanding stent [149] (Figure 3). The optimal valve geometry and leaflet thickness was generated after a series of in vitro testing. The resultant PHV for transcatheter surgery had a leaflet thickness of 230 µm and a distal stent approximately 50% larger in diameter than the proximal portion for fixation, with an effective orifice area of 1.38 cm^2^ (calculated using the Gorlin equation). The valve was crimpable to a 21F catheter [149].

The Polynova TAVR valve is the second-generation polymeric valve for transcatheter surgery made using xSIBS [150]. It is a modified version of the first-generation Polynova SAVR valve (Figure 3). The Polynova TAVR prototype valve was manufactured in-house using vacuum compression molding. The molding process consisted of placing raw xSIBS pellets inside the mold and then closing the mold under heat (220 °C) and pressure (4 Tons) for 1 h. The mold was designed to be connected to a vacuum line and hold vacuum throughout the molding process, therefore enhancing the density and quality of the molded polymer. The cured valves were then sutured to laser-cut nitinol stents (6-0 Silk black braided, Ethicon Inc., Raritan, NJ, USA) [150]. The leaflets of the valve partially mimic the optimized Polynova SAVR leaflet design, but the leaflets’ nominal confirmation (“zero-stress”) was adjusted to semi-open to further reduce the accumulated stresses over the cardiac cycle.

The Polynova TAVR underwent hydrodynamic testing with two benchmark tissue valves (Carpentier-Edwards PERIMOUNT Magna Ease SVAR valve size 19 mm (Edwards, Irvine, CA, USA) and Inovare TAVR valve size 20 mm (Braile Biomedica, Brazil). The polymeric TAVR valve had a larger EOA compared to the tissue valve. Additional hydrodynamic testing was performed using a patient-specific Replicator PD. The patient-specific anatomy was obtained from a patient with severe aortic stenosis. The largest EOA was observed in the Polynova TAVR valve. However, it demonstrated an average transvalvular pressure drop compared to the SAVR and Inovare valves. Despite better performance in systole, the Polynova TAVR valve showed the highest values of overall regurgitant fraction and required further optimization. The thrombogenicity values were lower in the polymeric TAVR and higher in the Inovare tissue valve. The obtained results indicated the necessity for the next design iteration to minimize regurgitation and optimize leaflet function during diastole [151].

Recently, Kovarovic et al. presented the second-generation polymeric TAVR valve [152]. The stent frame was optimized to generate larger radial forces with lower material volumes, securing robust deployment and anchoring. The leaflet shape, combined with varying leaflet thickness, was optimized for reducing the flexural cyclic stresses and the valve’s hydrodynamics. The optimization framework and technology led to the second generation of polymeric TAVR design currently undergoing in vitro hydrodynamic testing and following in vivo animal trials.

A new promising direction in the development of heart valves based on SIBS was the introduction of carbon nanofillers into the polymer matrix in order to improve the mechanical properties of the material, including resistance to cyclic loads [21,21]. Carbon nanotubes were introduced into the polymer structure by mixing solutions, and films were prepared by solution casting. The obtained composite structures had different properties, combining their own elasticity with the polymer and strength of carbon nanotubes. Nanocomposites containing SIBS and carbon nanotubes demonstrated a significant increase in tensile strength and elastic modulus compared to the original polymer [20]. The inclusion of a carbon nanofiller did not affect the biocompatibility of SIBS, showing no toxicity toward blood components like platelets and red blood cells [21].

Additional research involving hydrodynamic testing and animal testing is required to assess the future prospects of the project adequately.

### 3.5. PVA and PVA-BC

The Ontario group works with PVA, a hydrophilic polymer that is nonthrombogenic, has good biocompatibility and optimal mechanical properties beneficial for biomedical applications [153,154]. Linear PVA can be transformed into a solid hydrogel (cryogel) by physical crosslinking and using freeze–thaw cycles, thus creating materials that have properties similar to biological tissues [155]. The molecular weight of the raw material significantly affects the mechanical properties of the resultant polymer—the lower the molecular weight, the higher its rigidity [156]. The elasticity and strength of PVA may vary depending on the crosslinking conditions. The tensile properties of PVA depend on the number of freeze/thaw cycles. Its mechanical properties exhibit a stress–strain curve similar to porcine aortic root at the freeze and thaw cycle [157]. Nevertheless, in order to increase the strength and wear resistance of the hydrogel, the researchers attempted to further modify the properties of PVA: Mohammadi et al. reinforced PVA with bacterial cellulose (BC), a polyfunctional, hydrophilic, and biocompatible nanomaterial [124], to ensure that mechanical properties are mimicking the anisotropic mechanics of natural valve leaflets [157,158]. In this case, BC is a rigid element that creates and strengthens resistance to loads due to the fibrous structure. However, there are no studies of the long-term in vitro or in vivo functioning of this composite. Nevertheless, the authors have shown the possibility of using the material for the development of a TAVR prototype [159]. It is noteworthy that among the methods of increasing the biocompatibility of devices that come into contact with blood, such as cannulas, catheters, hemodialysis sets, and cardiopulmonary bypass systems, one can modify the surface of these devices by forming a hydrogel coating [160,161]. At the same time, polyacrylamide, poly(vinyl pyrrolidone), poly(methacrylic acid), as well as PVA can also be used as a material. The mechanism of biocompatibility increase is associated with an improvement in the hydrophilicity of the surface, i.e., a preventive decrease in the absorption of proteins on the surface, the blood coagulation, and the immune response [162,163]. It is shown that such modification is relevant primarily for short-term blood-contacting applications; however, presumably, PVA might be the most promising polymer for long-term blood-contacting PHV valves, taking into account the problems of cycle resistance.

Jiang et al. developed a novel tricuspid valve made entirely from polyvinyl alcohol cryogel (PVA-C) [154] (Figure 3). A cylindrical stent was produced from PVA-C, and its dimensions were replicated from the Medtronic Mosaic^®^ Aortic Bioprosthesis Model 305. The leaflet geometry was defined by a hyperbola, with the gap at the free edge region adjacent to the commissure areas. The central opening area of 5% of the stent orifice was chosen to ensure the closure of the leaflets. The stent has three posts to support the flexible leaflets. The sewing ring has three arc saddles to ensure conformity with the aortic annulus. The mechanical properties of the valve were successfully demonstrated by the reduction of flexural stresses in the leaflets. The stress concentrations at the commissural areas where the leaflets are attached to the stent were also avoided [154]. However, further studies are required to evaluate the potential of the PVA-C valve as a preferable alternative to the current valves. The PVA-C trileaflet valve was manufactured by cavity molding that formed the leaflets, sewing ring, and stent simultaneously. Hot PVA solution was injected into the mold, followed by water bath immersion and subsequent controlled freezing/thawing at −20 °C. This manufacturing approach ensured the maintenance of consistent properties of the material by avoiding disconnection of the leaflets and the stent at the commissure area [154]. Though the research group did not propose this valve prototype for transcatheter surgeries, after some modification, it might be suitable for it as well.

In developing TVR valve prototypes based on PVA-BC, the authors used an integrated approach based on numerical and experimental methods. The study involved modeling the pressure load on the leaflets made of a complex material model using ANSYS 15.0 software, taking into account both the properties of the “base” (Ogden model) and fiber hardening (Voigt model). Moreover, the authors used full-scale uniaxial stretching of PVA-BC samples to validate the underlying principles of anisotropy formation described in previous works of the team via freezing–thawing cycles in molds during pressure loading [164]. The geometry of the leaflet developed and manufactured on the basis of this approach was combined with a Nitinol basket and Dacron cover, forming a single-piece transcatheter heart valve. It is noteworthy that this model is presumed for the mitral valve, the primary replacement of which currently has no commercial implementation due to significant technical challenges such as providing a low profile with sufficient anchoring, preventing migration to the left ventricle, and a low risk of paravalvular regurgitation [165,166].

### 3.6. LLDPE and HA-LLDPE

The research group from Colorado State University focused on LLDPE, a biologically stable polymer made by copolymerizing ethylenes with long-chain olefins [127]. LLDPE has high tensile and tear strength, puncture resistance, low bending stiffness, and low-shear stress sensitivity enabling its application for the fabrication of cardiovascular devices [167]. However, neat LLDPE does not provide sufficient biocompatibility for prolonged blood–implant interaction, and as a result, researchers increase its hydrophilicity via combination with hyaluronic acid (HA) [126].

The introduction of hyaluronan into LLDPE by using the swelling method does not alter the mechanical properties but reduces the static contact angle [128,168]. The mean contact angle for HA-LLDPE film was 28.3 ± 20.0°, whereas for LLDPE—86.8 ± 4.2° [127]. HA enhancement significantly increases the hydrophilicity of the resultant material. None of the significant differences in cytotoxicity between HA-LLDPE and LLDPE were reported after 2 h contact with platelet-rich plasma [127].

Yousefi et al. tested several LLDPE valve prototypes to generate optimal design and leaflet configuration [93] (Figure 3). The stent consisted of two 3D-printed parts, with the bottom one securing the leaflets. The three stent posts form the three cusps of the valve. The rectangular piece of LLDPE is cut and wrapped around the bottom stent part, creating a cylinder. This piece is accurately fixed to the three stent posts to ensure a symmetric geometry in a closed position of the leaflets. The top stent part is placed on top of the leaflets in a way that creates three cusps resembling the trileaflet valve in the normally closed position. The top and bottom stent parts are secured together using three orthodontic rubber bands. The research group developed a higher valve profile and added leaflet arches that reduced Reynold’s shear stress (RSS), thus improving leaflet coaptation and minimal regurgitation percentage. However, a high stent profile may delay the reattachment of flow in the aorta and slightly increases RSS. Therefore, additional leaflet design optimization is required.

The LLDPE SVR valves were tested in an acrylic aortic chamber machined to mimic the outer walls of the aorta that was placed in the left heart simulator controlled by the in-house LabView software. Hemodynamic bench assessment reported an improvement in commissure coaptation based on the reduced back flow with either an increase in the profile length or the addition of leaflet arches. One prototype with a low profile and no leaflet arch failed during the test. Leaflet kinematics assessment indicates the increment in commissural contact as the aspect ratio and arch height increase. Quantitative flow assessment demonstrated similar flow patterns for all studied valve prototypes. The maximum velocity during early systole ranged from 1.3 m/s to 1.8 m/s for the LLDPE prototypes. Thus, a higher LLDPE profile drastically reduced the RSS values and helped to achieve better leaflet coaptation. However, this increment may provoke hemolysis and blood damage. Yousefi et al. concluded to optimize LLDPE valve geometry according to the obtained findings [93].

The HA-LLDPE transcatheter valve has a diameter of 26 mm, a height of 25 mm, and a balloon expandable cobalt chromium stent. The stent is comprised of two distinct rows of diamond-shaped structures and 3 “V” shaped structures connecting them. The top row is comprised of three larger diamond-shaped structures that represent the stent posts, whereas the connecting structures prevent interference of the native aortic leaflets with the polymeric leaflets. The leaflets are attached outside of the stent to fix their position during crimping [169].

The HA-LLDPE TVR valve was tested in an experimental pulse duplicator left heart flow simulator with two benchmark tissue valves, a 26 mm Medtronic Evolut (Medtronic, Minneapolis, MN, USA) and a 26 mm Edwards SAPIEN 3 (Edwards Lifesciences, Irvine, CA, USA). HA-LLDPE valve had EOA comparable to the Edwards SAPIEN 3 tissue valve (2.08 ± 0.04 cm^2^ vs. 2.10 ± 0.03 cm^2^, respectively), which was larger than that of the Evolut tissue valve (2.08 ± 0.04 cm^2^ vs. 1.80 ± 0.04 cm^2^, respectively). Heitkemper et al. supposed that the self-expanding principle of SAPIEN 3 contributed to its slightly larger EOA compared to the HA-LLDPE valve. The maximum velocity in the HA-LLDPE valve was 1.56 m/s at the acceleration phase, 1.94 m/s at peak systole, and 1.03 m/s at the deceleration phase, whereas Evolut and SAPIEN demonstrated 1.00 m/s and 0.86 m/s during acceleration, 2.45 m/s and 2.10 m/s at peak systole, and 1.37 m/s and 0.94 m/s at deceleration phase. The HA-LLDPE valve reported slightly decreased velocity at diastole as compared to Evolut and SAPIEN 3 (0.17 m/s vs. 0.19 m/s vs. 0.19 m/s, respectively). Thus, a regurgitant fraction of the HA-LLDPE valve was comparable to both benchmark valves. Interestingly, Heitkemper compared the regurgitant fractions and EOA between the HA-LLDPE and TRISKELE valves based on available literature data and found that the regurgitant fraction was lower in the HA-LLDPE valve, whereas its EOA was larger [169]. Velocity and vorticity analyses revealed that the HA-LLDPE valve had improved turbulent flow characteristics compared to the commercially available controls [169]. Based on the reported bench assessment, the HA-LLDPE valve`s hemodynamic performance and turbulence were comparable to Evolut and SAPIEN 3, and it may be regarded as an alternative to the widely used TVR tissue valves.

### 3.7. Trends in the Development of PHV Materials

After more than 50 years of research and development, recent studies are mainly focused on composite and hybrid materials capable of mimicking the structure of the native leaflet. The most promising materials combine components that provide leaflets with elasticity, high strength, and resistance to cyclic loads. New polymers are modified original polymers (e.g., polyurethanes and silicones) with improved properties using unique polymeric blends, nanofillers, and composites. Along with the development of surgical PHV, polymeric TAVR valves are highly demanded since novel polymers possess unique properties that mimic native leaflets. They are thin but durable and resistant to cyclic load and may retain their functional properties after setting to the balloon and expansion during the implantation. The efforts of researchers are aimed at improving hemocompatibility by applying different coatings or changing the polymer structure itself.

## 4. Challenges and Future Prospects

Previously published reviews do not cover issues ranging from the polymer composition and structure up to in vivo testing, pre- and clinical trials, as well as potential drawbacks and approaches to overcome the existing limitations. Most of them are addressed to specific topics detailing mechanical properties or perspectives of their application for transcatheter technologies. This section presents key technologies, ideas, and trends in the development of superior materials for flexible leaflets of an elastic polymeric heart valve.

Despite the general appeal of the TEHV valves, this technology has not found a clinical application yet [6,170] since the first implantation in sheep in 1995 [171]. The development of PHVs with biostable elastic leaflets remains one of the promising alternatives to modern prosthetic heart valves made of biological materials and rigid synthetic polymers. This direction is gaining traction nowadays due to the recent progress in the development of new composite polymer materials, which still possess poor resistance to cyclic loading and hemocompatibility. At the same time, the development of composites that mimic the anisotropic and multilayer structure of native valve leaflets, where each layer (*n* = 3) performs a specific function, can be a solution to this problem (Figure 4).

One of the ways to enrich anisotropic elastic materials resistant to cyclic loading is to assemble microdomains of block copolymers with cylindrical morphology that are able to align with mechanical shear in accordance with the design features of the heart valve leaflets [146]. Cylindrical microdomains of block copolymers based on end-blocks of styrene with an elastic block of isoprene, isobutylene, butadiene, or ethylene–butylene can reach the size of tens of nanometers [96]. The bidirectional orientation can be achieved by micro-injection molding and conventional injection molding. This approach was used by Stasiak and colleagues to obtain the heart valve with anisotropic domains forming a layered structure exhibiting bi-directional orientation. Numerical modeling made it possible to explain and predict the complex orientation of microstructures in the material by the balance of shear and extensional flow [146,172].

Fiber-reinforced polymer composites with carbon nanofibers or carbon nanotubes provide an opportunity to obtain anisotropic valve prosthesis use [173,174]. Similarly to SIBS reinforced with Dacron, the fibers of a stronger polymer introduced at the macro level do not provide the desired resistance to plastic deformation under cyclic loading since the polymer base significantly affects the results [149]. At the same time, carbon nanotubes significantly improve the mechanical properties of the polymer matrix in terms of the development of the heart valve leaflets by changing the nano- and microstructure [20,21]. However, without the application of special alignment methods, the properties of nanocomposites change isotropically because the nanofillers are distributed randomly. Most known alignment methods involve the use of external forces, such as electric or magnetic fields, shear force, or mechanical stretching. Alignment can be achieved using “ex situ” or “in situ” methods before or after the introduction of polymers [175].

Over the past decades, electrospinning has been widely used in the tissue engineering of blood vessels, providing a porous fibrous structure similar to native tissues and, under certain conditions, achieving anisotropy of mechanical properties of matrices [176,177]. Combined microfabrication and electrospinning eased the development of three-layer biomimetic scaffolds for tissue-engineered vascular grafts [177]. Biodegradable structures made on the basis of poly(glycerolsebacinate) and poly(caprolactone) functioned well in the porcine ex vivo model of heart valve leaflet replacement. A similar model of biomimetic multilayer polymer materials intended for the repair and replacement of heart valves was studied by Sun et al. [176]. Porous polycarbonate urethane was selected as a polymer that would imitate the spongy layer of the native valve. Polycaprolactone fabricated by hybrid electrospinning with highly oriented fibers and solution casting has become an alternative to fibrosa and ventricular layers [176]. The obtained composite material demonstrated anisotropic mechanical behavior and rigidity similar to native aortic valve leaflets, as compared to three different commercially available patches.

Moreover, complex anisotropic scaffolds can be formed using high-precision 3D printing [178], which allows for setting different orientations, thicknesses, tortuosity, and porosity of layers in the leaflets [179]. High-precision printing systems are capable of completing multi-material printing with a resolution of up to 0.02 mm (20 microns), whereas CAD/CAM automation can enable and improve their design [179]. However, the polymers that could be used for such purposes are not usually tested in that regard, and thus it is possible that they lose stability or degrade due to thermal conditions (FDM, SLS printing) or exposure to UV curing (SLA printing) [180].

Numerical modeling and design allow the modeling of composite fiber materials using the finite element method. They ensure the calculation of the optimal filler–base ratio and fiber orientation even before the receipt of the composite samples. Some modeling software (CAE), like Abaqus Multimech, partially implements this approach via sub-modeling techniques [181], but they require validation and a deeper understanding of native tissue micromechanics for materials to be reproduced in silico. Machine learning can be considered one of the promising approaches to model composites and generate semi-automatically new models of materials, excluding engineer bias in the final topology that sometimes results in polymers with nontrivial structures. This approach, based on neural networks, has shown success in creating complex architectural materials and composites [182,183], which can later be translated to PHV development. The combination of complex modeling and generative algorithms describing and predicting the behavior of such materials can contribute to significant improvements in the field of multicomponent polymeric materials, especially in those containing two or more types of fibers in multilayer systems.

Generally, there are a number of potential directions for PHV development and application, whereas the results of clinical trials of the Tria valve will define the new era of polymeric heart valves.

## 5. Conclusions

MEDLINE, Scopus, Clinicaltrials.gov, Google Scholar, and Web of Science databases were used to identify research articles published between 2000 and 2022. However, some individual studies of lesser-known polymers in the early stages of their development or of well-known polymers intended for other medical applications may have been overlooked. Nevertheless, the review accumulates data on all recently emerged and known polymers that are currently used for fabricating polymeric heart valves.

Currently, the problem of structural valve degeneration can be solved only by using either mechanical or biological prostheses. Despite the wide variety of materials and manufacturing technologies discussed in the review, the advancements in this field are limited by the strict requirements for mechanical properties and biostability, the complex and anisotropic behavior of the valve model, and the lack of self-healing synthetic materials. To date, LifePolymer (Foldax) is the only biopolymer material that has successfully passed preclinical tests and has been implanted in humans during clinical trials. However, in light of recent advances in high-molecular compounds and materials science, especially in the development of various copolymers, nanocomposites, and other hybrid structures combining the advantages of compounds, it has become possible to develop patient-specific anisotropic heart valves. New additive manufacturing technologies, such as 3D printing, electrospinning, and microfabrication technologies, have brought the global biomedical community closer to the development of an optimal heart valve prosthesis.

## Figures and Tables

**Figure 1 ijms-24-03963-f001:**
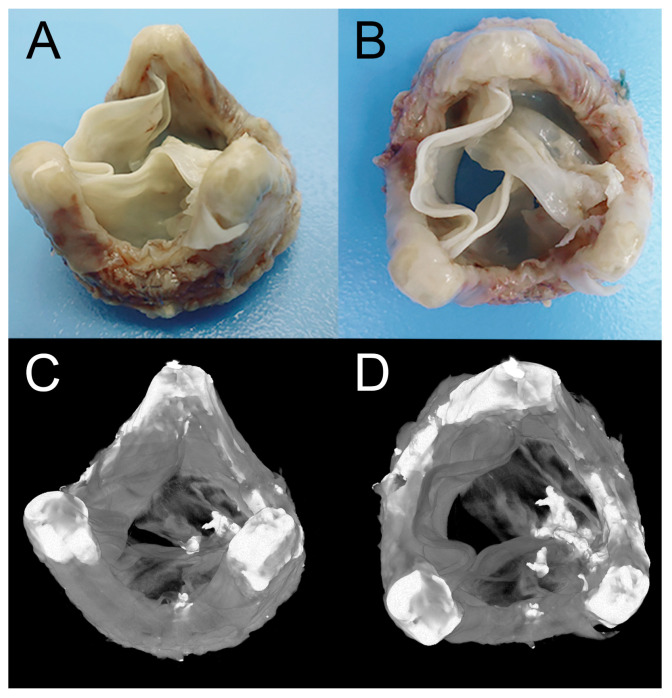
Explanted biological heart valve with degenerative changes (signs of fibrosis and calcification), (**A**)—outflow portion (isometric view), (**B**)—outflow portion (top view), (**C**)—microtomographic projection of outflow side (isometric view), (**D**)—microtomographic projections of outflow side (top view).

**Figure 2 ijms-24-03963-f002:**
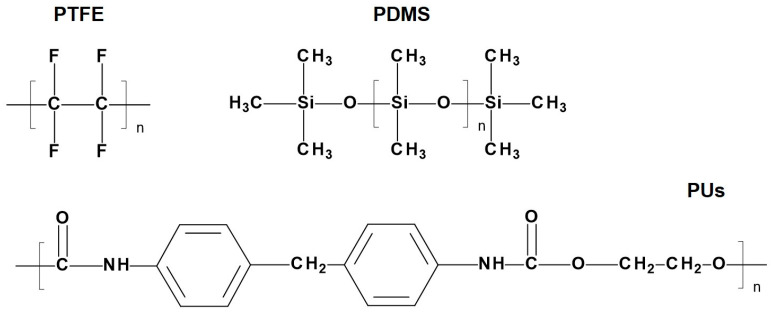
Structural formulas of materials for the first-generation heart valves.

**Figure 3 ijms-24-03963-f003:**
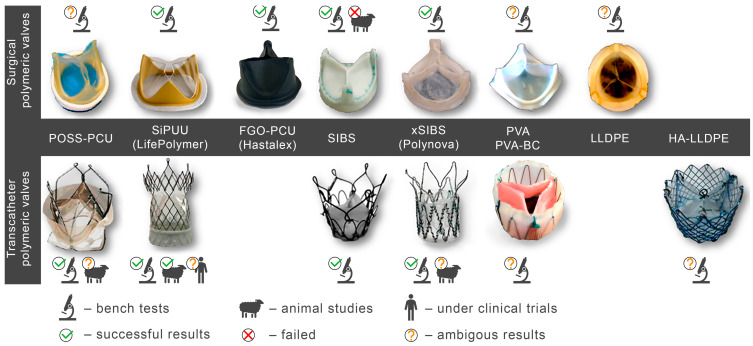
Polymeric valves developed in the last decade for surgical open-heart valve replacement (SVR) and polymeric valves for transcatheter valve replacement (TVR).

**Figure 4 ijms-24-03963-f004:**
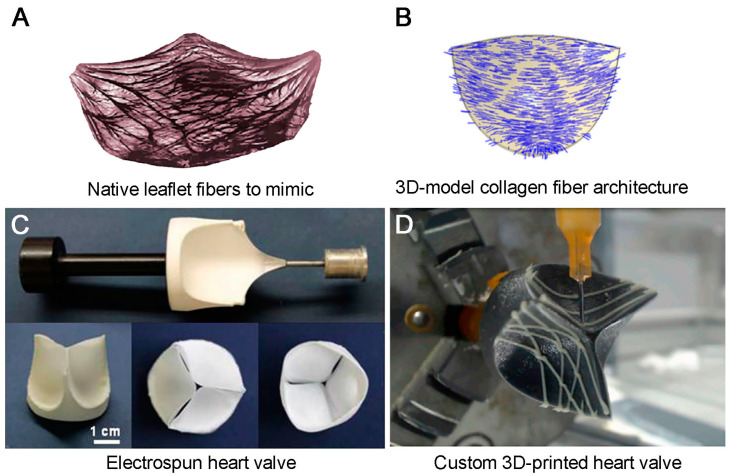
Orientation of collagen fibers in the native leaflets (**A**) (Coulter, 2019); fiber orientation modeling in numerical modeling (**B**) (Serrani, 2016); prototyping and electrospinning of a polymeric heart valve (**C**) (D’Amore, 2018); prototyping of a leaflet using a modified 3D printing system (**D**) (Coulter, 2019).

**Table 1 ijms-24-03963-t001:** Physical and mechanical properties of novel polymeric materials.

Biomaterial	Chemical Structure	Mechanical Properties	Surface Properties
POSS-PCU [114,115,116]	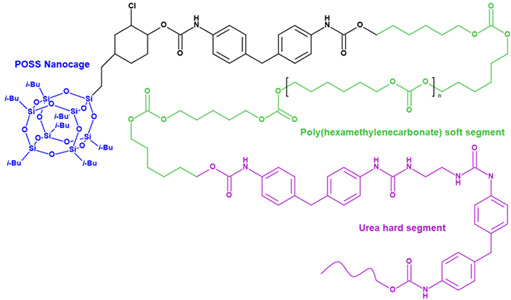	Tensile strength of 53.6 ± 3.4 MPa Elongation at break of 704.8 ± 38.0% Young’s modulus of 25.9 ± 1.9 MPa Tear strength of 50.0 ± 1.2 MPa	Contact angle of 100.3° ± 2.7°
SiPUU [117,118]	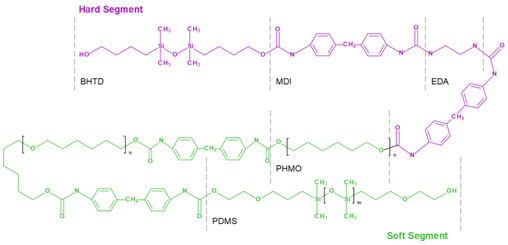	Tensile strength of 31.0 ± 2.4 MPa Elongation at break of 646 ± 24% Young’s modulus of 18.0 ± 0.7 MPa Tear strength of 64.0 ± 2.3 N/mm	Contact angle of 113.6° ± 0.9°
SIBS [119,120,121]	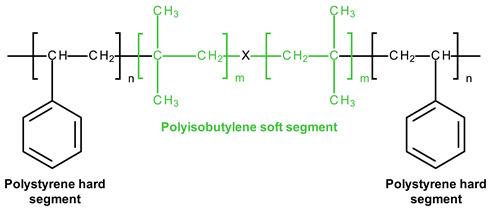	Tensile strength of 3.70 ± 0.31 MPa Elongation at break of 384.70 ± 20.78 Young’s modulus of 4.08 ± 1.17 MPa	Contact angle of 72.3° ± 3.0°
xSIBS [122]	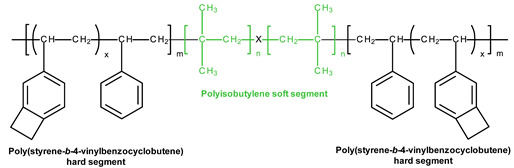	Ultimate tensile strength of almost 35 MPa	Contact angle of 82.15° ± 0.02°
PVA-C [123]	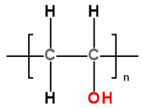	Tensile strength of 37.3 MPa Elongation at break of 165.9%	Contact angle of 40.0° ± 2.4°
PVA-BC [123,124,125]	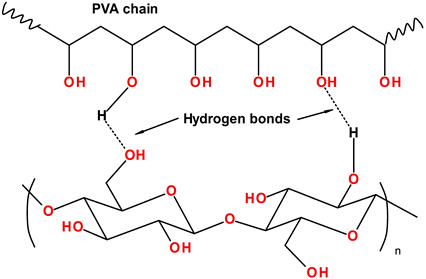	Tensile strength of 60.9–74.5 MPa Elongation at break of 9.6–13.8%	Contact angle data are not available
LLDPE [126]	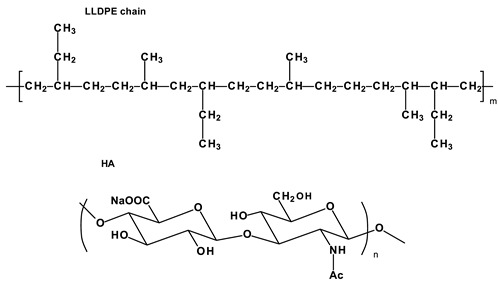	Yield strength of 7.29 ± 0.29 Bending stiffness of 26.10 ± 3.62 Elongation at break of 582 ± 23% Young’s modulus of 73.82 ± 6.83 MPa	Contact angle of 86.8° ± 4.2°
HA-LLDPE [126,127,128]		Yield strength of 8.23–9.74 MPa Bending stiffness of 12.93–21.72 Elongation at break of 476–787% Young’s modulus of 76.49–99.71 MPa	Contact angle of 45°
FGO-PCU [26]	not available	Tensile strength of 57.1 MPa, Elongation at break of 1004.3% Young’s modulus of 11.3 MPa	Contact angles of 106.4° ± 0.1° for the shiny surface and 85.2° ± 1.1° for the opaque surface

## Data Availability

No additional data.

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
