# Peer review of "Polymeric Heart Valves Will Displace Mechanical and Tissue Heart Valves: A New Era for the Medical Devices"

_ijms, 2023, doi:10.3390/ijms24043963_

Round 1
Reviewer 1 Report
1. As your abstract's final sentence, include a "take-home" message.
2. Put the keywords in a new order based on alphabetical order.
3. In line 2-3 of title, “a Brave New World”, what the specific reasons for that? The reviewer thik it would be suitable if changes to “A review” or “A narrative review”. Any explanation or rebut?
4. The Reviewer do not see the novel in the present review. My examination revealed that several similar previous publications appear to appropriately address the issues you have brought up in the current submission. Please emphasize it more advance in the introduction section if there are any more truly something really new.
5. In order to highlight the gaps in the literature that the most recent research aims to fill, it is crucial to review the benefits, novelty, and limitations of earlier review literature in the introduction.
6. Before line 973, The limitation of the present review needs to be added before entering the conclusion section.
7. In line 955-968, the authors have been explained finite element analysis as in silico study for further research. Please explain that in silico study have been several advantages in terms of faster results and lower cost to investigate polymeric implant materials compared to in vivo and in vitro. Additional MDPI reference needs to adopted as follows: Computational Contact Pressure Prediction of CoCrMo, SS 316L and Ti6Al4V Femoral Head against UHMWPE Acetabular Cup under Gait Cycle. J. Funct. Biomater. 2022, 13, 64. https://doi.org/10.3390/jfb13020064
8. The reference needs to be enriched from the literature published five years back. MDPI reference is strongly recommended.
9. The manuscript needs to be proofread by the authors since it has grammatical and language issues.
10. It is suggested to the authors for providing graphical abstract in the system after revision.
Reviewer 2 Report
The review paper by Rezvova et al. presents and analyzes different aspects related to polymeric heart valves including materials (of different generations, e.g. monocomponent, multicomponent and nanotechology-based) and their properties, fabrication methods and effects related to selecting each of them, as well as outlook and expectations on the further development of the technology. Overall I find the article comprehensive and of interest for the journal readership. There are some parts of this work that can be improved in terms of structure and clarity, as detailed in the comment list below:
- Line 38: Please specify which timeframe the number 250000-300000 is relative to.
- Line 60-61: I don't understand "their obvious necessitate", did you mean "they obviously necessitate"?
- Lines 108-110: The following sentence seem to be from the template? Please remove/clean up as appropriate: "This section may be divided by subheadings. It should provide a concise and precise description of the experimental results, their interpretation, as well as the experimental conclusions that can be drawn".
- Line 341: Please add the specific reference number.
- Lines 406-422 These lines contain a conclusion on ideal characteristics based on the challenges and phenomena presented in section 2.2, so I recommend that they are part of a separate section for clarity (e.g. it could be section 2.2.6).
- Table 1, page 10, POSS-PCU: Mismatch in significant digits for elongation at break. Please correct.
- Table 1, page 10, SiPUU: Mismatch in significant digits for Tensile strength, Young's modulus, Tear strength. Please correct.
- Table 1, page 11, PVA-C: Last column says "not available". Do you refer to "Contact angle"? Please specify in the table.
- Line 553: Do you mean ISO 10993 series of standards?
- Line 818: You probably meant an "and" instead of the "u".
- Line 828: Mismatch in significant digits for the contact angle (average and standard deviation). Please correct.
- Line 870: Mismatch in significant digits (1.8 ± 0.04 cm2). Please correct.
- Lines 887-892: Those lines wrap up on the previous sections 3.x, so I recommend that they are part of a short, separate section instead (e.g. it could be section 3.7).
Round 2
Reviewer 1 Report
Good work to the authors, several comments as response in their revision is given:
1. The urgent reasons of using polymers based material for hearth valve needs to be highlighted.
2. The authors needs to giving additional data related to mechanical properties of established polymeric hearth valve.
3. In line 944-966, the authors explain the scaffold. The authors encouraged to explain the role of scaffold in tissue engineering for improve the explanation in this section. Suggested reverence needs to be adopted as follows: The Effect of Tortuosity on Permeability of Porous Scaffold. Biomedicines 2023, 11, 427. https://doi.org/10.3390/biomedicines11020427
4. Additional figure in section 1. Introduction would increase the presentation quality of present article.
Round 3
Reviewer 1 Report
The present work is recommended to publish in IJMS.